# Efforts of a Mobile Geriatric Team from a Next-of-Kin Perspective: A Phenomenographic Study

**DOI:** 10.3390/healthcare11071038

**Published:** 2023-04-04

**Authors:** Kjerstin Larsson, Veronika Wallroth, Agneta Schröder

**Affiliations:** 1University Health Care Research Center, Faculty of Medicine and Health, Örebro University, 702 81 Örebro, Sweden; 2Division of Social Work (SOCARB), Department of Culture and Society (IKOS), Linköping University, 581 83 Linköping, Sweden; 3Department of Nursing, Faculty of Health, Care and Nursing, Norwegian University of Science and Technology (NTNU), 2815 Gjövik, Norway

**Keywords:** mobile geriatric team (MGT), next of kin, older adults, phenomenography

## Abstract

Many older adults with complex illnesses are today cared for by their next of kin in their own homes and are often sent between different caregivers in public healthcare. Mobile Geriatric Teams (MGTs) are a healthcare initiative for older adults with extensive care needs living at home, coordinated between hospital, primary, and municipal care. The study aims to describe how next of kin experience care efforts from an MGT for their older adult family members. The study has a descriptive qualitative design and uses a phenomenographic approach. Fourteen next of kin to older adult family members who receive efforts from an MGT were interviewed. Two descriptive categories reflecting their experiences emerged: Professional care and No longer having the main responsibility. The study shows that the participants valued that the staff was very competent, that the physician made home visits and could make quick decisions, and that treatments were given at home. They feel that they receive support and experience security and that a burden is lifted from them. Our study shows that through the MGT, next of kin become involved in the care and are relieved of the burden of responsibility of caring for their older family member.

## 1. Introduction

In Sweden, as in many parts of the world with growing numbers of older adults, elder care services are facing both immediate and long-term challenges. The proportion of older people in the population is increasing [1], where “older” often refers to the ages of 65 and over in Sweden. Today, neither public financial resources nor public elder care services cover the existing needs. Contrary to many people’s beliefs, it is not public care for older adults that provide the most care in Sweden. Instead, the next of kin of those in need provide most of the care [2]. Generally, people who care for older adults are also older themselves [3]. Usually, spouses/partners care for each other, and this group of caregivers has grown in recent years. According to Jegermalm [3], the percentage of households providing family care in their own home has increased from 5 per cent in 2005 to 7 per cent in 2019. Of these, 34 per cent help someone who is 75 years or older. Helping someone in need of care living at home is common among both women and men, but the next of kin who provide care in their own household provide significantly more hours per month (66 h) than those who help someone outside their own household (15 h) [3]. In other words, much of the more comprehensive care for older adults is provided by a spouse or partner in their joint household, and research in this area indicates that this type of caring responsibility has increased over time [4].

With more and more older adults in need of care and more and more next of kin who provide it, next of kin today also provides care to older adults with complex disease pictures. It is also a well-known problem that older adult patients often are sent back and forth between different care providers [5]. This causes not only stress for the older adults themselves but also for their next of kin who must accompany them, requiring more effective support from the next of kin and increasing demands on collaboration [6]. Within geriatric healthcare, there are several examples of collaboration and multidisciplinary teams working together to provide holistic home-based healthcare. The most used method for holistic acute care of older adults is the Comprehensive Geriatric Assessment (CGA) (cf. [7]). CGA is defined by Rubenstein, Stuck, Siu and Wieland [8] as a “multidimensional interdisciplinary diagnostic process focused on determining a frail older person’s medical, psychological and functional capability in order to develop a coordinated and integrated plan for treatment and long-term follow-up” (p. 1). Also, Ellis and Sedvalis [9] write that the “gold standard” of care for older adults is CGA. They emphasise that CGA has evolved over several decades and across continents.

The Mobile Geriatric Team’s (MGT) ambition is to take a holistic approach and see the person in need of care as a whole, considering medical, social, psychological and physical aspects, as well as taking the perspective of the next of kin into account. This is in agreement with the National Board of Health and Welfare’s [10] definition of a holistic view. One example of collaboration from a holistic perspective is given by Kohn, Goldsmith and Sedgwick [11], who evaluate the general feasibility of using a multidisciplinary psychiatric mobile team for older adult patients. Their results show that the team appeared to lessen psychiatric disability for older adults with mental illness. Mi Oh, Warnes and Bath [12] evaluated the effectiveness of a rapid response service for frail older adults in Barnsley, UK. They concluded that the rapid response service provided a holistic assessment where the team identified and responded to several needs that regular healthcare had not perceived and grasped. In another UK study, Clift [13] explored the work of a rapid response team in Southampton. A conclusion from the study is that the team can manage patients safely in their own homes during clinical crises, avoiding unnecessary and often detrimental lengthy stays in acute hospitals. In addition, a Swedish study by Westgård, Wilhelmson, Dahlin-Ivanoff and Ottenvall Hammar [14] focusing on frail older people’s experiences in an acute geriatric ward practising CGA concluded that feeling “respected as a person” was important for the patients. This was achieved by having a reciprocal relationship with the ward staff, enabling the patients to participate in decisions when engaged in communication and understanding. Swedish researchers Fristedt, Nystedt and Skogar [15] write that different forms of MGTs have become a more common form of collaboration in Sweden, as well as internationally. However, scientific evaluation of this initiative has been scarce [15]. In their randomised controlled trial examining MGTs, Fristedt et al. [15] focused on cost-effectiveness and user satisfaction. The results show that the MGT initiative was clearly appreciated and highly valued by the users, i.e., the older adults. However, the initiative did not have the desired effect in terms of cost efficiency, and the older adults’ consumption of hospital-based healthcare did not decrease compared to the control group. They conclude that more follow-ups on MGT interventions are needed. As shown in a Swedish study by Wallroth [16], family caregivers often find themselves advocating, defending and claiming their older family members’ rights at different health and caregiving institutions. When professional care and medical institutions fall short, or when the next of kin do not trust those who provide the care or medical efforts, it causes a lot of anxiety and stress. In another Swedish study focusing on security and quality of care, Törnfelt, Roos and Hellström [17] explored the view of next of kin on acute mobile geriatric care for older people in their own homes. The results show that the next of kin appreciated interpersonal factors, such as good communication, empathy, having a reduced number of contacts, and the mobile team taking time for the older adult and being easy to reach. These factors were emphasised as creating a sense of security and quality of care among the older adults’ next of kin. However, the authors also conclude, in line with Fristedt et al. [15], that more studies are needed on the next of kin’s views on MGTs. To our knowledge, there are no previous Swedish studies that have focused on the next of kin’s experiences of MGT efforts. Therefore, the aim of this study is to describe how next of kin experience care efforts from a mobile geriatric team for their older adult family members.

## 2. Materials and Methods

### 2.1. Setting—The Mobile Geriatric Team

The “ÄDEL” reform in Sweden in the 1990s contributed to a separation of medical care managed by the region and that managed by the municipality. During the 1990s, the municipalities were given the responsibility for all medical care and welfare services at care homes and local nursing homes, with the exception of medical interventions, which continued to be the county council’s responsibility [18]. Between the years 2010 and 2014, a Swedish governmental initiative was introduced with the purpose of improving the quality of life for “the sickest older persons” through coordinated care between hospital-based care and primary care, which also includes municipal care. Since 2017, this type of coordination has been articulated in the Act on Collaboration in Discharge from Hospital Care [19]. The government has decided on a reform of primary care in Sweden under the motto of “good and nearby care”, with a special focus on accessibility, participation and continuity. The goal is for the patient to receive good, nearby care that strengthens their health [20]. This reform is a response to a healthcare system that has become too divided and specialised, and the lack of coordination has led to patients falling through the system [21]. Lack of coordination is also something that is highlighted in the recently adopted National Next of Kin Strategy within health and social care [22].

As part of these initiatives, reforms, and legislation, several mobile geriatric teams were created by Swedish health and social care authorities. The purpose of these teams is to accomplish Comprehensive Geriatric Assessments in the patient’s home [15,17]. The MGT at focus in this study consists of two physicians working part-time in the project, one care administrator working part-time, and five full-time nurses. The MGT also works in close collaboration with a counsellor, an occupational therapist, and a physiotherapist and has a strong collaboration with the municipality’s regular home care staff. At the time the study was conducted, the team had the capacity to care for 20–25 enrolled older adults. Another idea behind the MGT is that the advanced home-based healthcare provided by the team should reduce the number of visits to the emergency room, which older adults otherwise often have to make. In addition to the inconvenience of getting to the emergency room when one is very ill, these visits often entail a long wait before help is received. This also means that the disease picture around the older person becomes fragmented. The MGT can easily, and without a referral procedure, contact specialists in the region’s somatic and psychiatric clinics for advice and support.

### 2.2. Design

This study has a descriptive qualitative design. A phenomenographic approach was used to investigate the qualitative variations in how people perceive, experience, conceptualise and understand phenomena in the world around them [23]. While individual conceptions are central in phenomenography, the result is a description on the collective level in the form of distinct, descriptive categories that capture the variation among and within individuals [23,24]. Phenomenography makes a distinction between what something is, “the first-order perspective”, and how something is perceived to be, “the second-order perspective” [25]. The second-order perspective is central in phenomenography [23], meaning that the main interest lies in how people perceive the world rather than what it “really” is like.

### 2.3. Participants

The main selection criterion for the study was that the participants had to be next of kin to an older adult who has contact with the MGT. A contact person within the MGT was designated and selected as the next of kin for this study based upon selection criteria provided by the research team, namely that there should be variation among the participants with regard to gender, age and their relationship to the older adult. Another criterion was that the health status of the next of kin could vary as well so that some next of kin could be in good health while others could be in need of healthcare themselves. Fifteen next of kin were asked to participate in this study, of which one declined. The sample, thus, came to consist of fourteen next of kin to older adults aged 36–82 years (see Table 1). The interviews were conducted about three to four weeks after the older adults had been enrolled in the MGT project. 

### 2.4. Data Collection

Data were collected during the spring of 2019 using individual interviews carried out in the next of kin’s home, their workplace or the researcher’s workplace. The initial question was: “How do you, as next of kin, experience the mobile geriatric team’s healthcare efforts?” Follow-up questions were used to capture variations in the conceptions, for example: “Can you explain?”, “Can you tell me more?”, “What do you mean?” or “Is there anything else you want to say, something I have not asked you about?” The semi-structured interviews, in line with the phenomenographic approach [26], were performed by one of the authors (KL), a social worker with experience in qualitative interviews. The interviews lasted between one and one and a half hours and were audio-taped and transcribed verbatim. All data collection was completed before the analysis started. 

### 2.5. Data Analysis

Phenomenographic analysis is directed toward similarities and differences between individual statements and moves from an individual to a collective awareness as a “pool of meaning” [23]. The analysis was conducted manually to allow greater intimacy and familiarity with the data. The analysis was performed in four steps in accordance with Marton [27] and Marton, Dall’Alba and Beaty [28].

In the first step, the recording was listened to in order to make sure that the transcribed text was correctly transcribed and to detect any possible inaccuracies. Thereafter the whole text was read several times with an open mind to become familiar with the data and identify statements relevant to the aim of the study. Following these readings, notes were taken.

In the second step, similarities and differences between the ways the participants described the phenomenon were identified. Distinct statements were labelled, and from these labels, preliminary conceptions were formed.

In the third phase, the conceptions were compared and grouped into preliminary descriptive categories in order to obtain an overall map of how similarities and differences could be linked.

In the fourth phase, the relations between the preliminary descriptive categories were in focus [27]. All descriptive categories were critically scrutinised to ensure that they were in agreement with the conceptions and distinguished from one another. Finally, two descriptive categories emerged, empirically based on the interviews. 

There was a constant interplay between the various steps of the analysis, and we had thorough discussions using an iterative approach as described by Bruce et al. [29]. The findings are presented in an outcome space with a horizontal structure [30], as seen in Table 2.

### 2.6. Ethical Considerations

This project follows the Swedish Research Council’s principles for research ethics and the guidelines for good research practice [31], as well as the European Code of Conduct for Research Integrity [32]. The project also follows the General Data Protection Regulation (GDPR) regarding the processing of personal data. In the recruitment process, an information letter was sent to the participants explaining the purpose and approach of the study. A consent form was sent along with the information letter. Before the interview started, the participants received information about the aim of the study and filled out the consent form. This means, among other things, that participants were informed about the purpose of the study, that they gave their consent to participate, and that they were informed that they could withdraw their participation at any time before the results were published. Recorded interviews, transcribed material and any information that can reveal the identity of the participants are stored and handled in such a way that unauthorised persons cannot obtain the material. Furthermore, the interviewees were informed that all information that emerged during the interviews would be handled with the greatest possible confidentiality. All participants were legally competent to give their consent. The study was approved by the Regional Ethics Committee in Uppsala (diary number: 18RS4491).

## 3. Results

The two descriptive categories that emerged in the analysis were: Professional care and No longer having the main responsibility, which includes six different conceptions (see Table 2). In the presentation of the findings, the conceptions are illustrated with quotations from the interviews. 

### 3.1. Professional Care

This descriptive category brings together the three conceptions of Competent staff, Person-centred care and Increased availability.

#### 3.1.1. Competent Staff

The next of kin felt that there was not enough competence in municipal care and that there was a lack of competence regarding issues concerning older people in primary care. They expressed that they thought it was very good that the people on the MGT who carried out the examinations were competent. Most of the next of kin felt that there was a difference in competence between the regular home healthcare and the MGT, as the MGT staff were handpicked and had more specialist competence.

The next of kin also experienced that the MGT had more direct contact with specialists within the hospital that they could consult with or send the older person in need of care to without a lot of waiting time. Having a physician on the MGT also meant that the team had the authority to make decisions much faster. 

They [the MGT] have fulfilled all criteria. They have been professional on all fronts. I felt really secure when they took over. I knew she would get the best care because when we had the first meeting, I gained confidence in them immediately. And then it was also, with the result, all the time things happened that were better. You could see an improvement. When you see an improvement, well then there is not so much to babble on about, so to speak.(Hans)

#### 3.1.2. Person-Centred Care

Several of the next of kin felt that previously, no one had taken overall responsibility for the patients when they visited health centres, emergency rooms or wards in the healthcare system. Rather, the next of kin’s experience was that these institutions made referrals to each other without anyone taking on the overall responsibility. Most of the next of kin described the importance of the MGT taking an overall responsibility and, thus, a holistic approach to the patient’s situation, as older people with complex care needs often have multiple problems, which means that they are usually referred to different specialists. The MGT investigates all the patient’s problems and takes responsibility for them as a whole. They get to know the patient and next of kin, creating a sense of security for both of them.

I think the efforts look at the whole picture, that is, both medically, clinically and in care. … [Before, my mother] could not get intravenous fluid, or could not get any extra blood if needed. Nursing these wounds and so on, we did not have, it would not have worked.(Daniel)

Most next of kin described extensive drug treatments that had not been coordinated before. When the MGT was involved, the team’s physician started to investigate all the drugs and reduced several of them with the help of the clinical pharmacist. Thus, several next of kin described that because a holistic view is missing in primary care, drugs are being prescribed without much thought and without any evaluation of whether the medicine is helping or risks working poorly with other medicines or whether it should no longer be taken.

Several of the older adults were described as having multiple problems that are often related to one another. Several next of kin found that it was difficult to be involved and ask questions all the time during visits to the health centre and the emergency room. The next of kin, thus, described a rather fragmented care system where no one takes the time to obtain the whole picture. When the MGT came into their lives, they offered continuity, and this was a huge difference from how they used to be treated by medical authorities. The next of kin appreciated that the MGT physicians made home visits and that it always was the same physician they had contact with. Furthermore, the next of kin thought that when you have continuous contact with the same people, you become familiar with them, which is very important in all respects.

The doctors came and … the nurses also come and look, with a little more continuity … a little more often, to find these paths. What medications should we change? What medications should we replace? And what should we conclude so that it will be best? And it’s like that, it’s the workflow I like, because then it only took a month and then she felt fine. (Hans)

#### 3.1.3. Increased Availability

The next of kin appreciated that they quickly could obtain help from the staff in the MGT and expressed that there is a big difference compared to getting help via the health centre. The next of kin described that the physicians and nurses come to the older person’s home and begin the investigation and assess the patient’s problems. They expressed that there was an incredible difference in quality for the next of kin and the older adult when the physician visited them at home. The next of kin appreciated that the MGT physician made home visits and that it always was the same physician they had contact with. 

I think the team can do other things. They can give intravenous antibiotics. We have, according to the agreement with the municipality and with primary care, there are certain things that they do not do, they do not give blood transfusions [in the older adult’s home] for example, they do not give intravenous antibiotics. The team can do that, and such measures can really increase the quality of life for those who are seriously ill. Not having to go to a health centre or be admitted or …. (Helena)

All next of kin experienced that they had gained access to wonderful healthcare through the MGT. They highly appreciated that their loved ones were examined and assessed at home and that they had close contact with the team. The team can follow the development of the treatment and can quickly implement measures in the event of deterioration. In the past, older adults often had to wait a long time before receiving help, which led to unnecessary complications and stress for the next of kin. All the next of kin described that the number of visits to the emergency room so far had been reduced to none since their older family members had started receiving medical efforts from the MGT.

### 3.2. No Longer Having the Main Responsibility

This descriptive category brings together three conceptions: Trust, Inclusion and Relief.

#### 3.2.1. Trust

The next of kin described how their worries about their relative had lessened and that they felt a great sense of security now that the team had the responsibility for the care and nursing of their next of kin. They believed that their everyday life had become calmer and safer when the team got involved. The next of kin described that they had confidence and trust in the team. The MGT staff investigate all the patient’s problems and takes on the overall responsibility.

Yes, since they came in [The MGT] I’ve definitely been much calmer. [I have] great confidence in what has happened. Thanks to seeing improvements again. I don’t have to worry anymore. So, a sense of security. A calm, you could say.(Hans)

Some next of kin said that they had had a tough situation before and that they now could see a change in their lives, from constantly feeling anxious and responsible for their loved ones to being able to feel safe. They experienced that the team’s efforts were of high quality and gave them a bit of freedom.

It has been safe. I have been able to feel calm. I know that she is being taken care of in the best way, and this knowledge means that I can relax and don’t have to worry and spend energy and think of how to solve the situation so that she will have the best possible time. (Valter)

It offered a sense of security, the next of kin said. They described that what is important is proximity, accessibility, top competence and the team solving the problems faster than other healthcare. This means that the next of kin felt confident with the MGT. 

Then the responsibility rested on me, you could almost say … If he was to sound the alarm or call or if he got sick or if he could be at home. Now they are the ones who are responsible for him. … So, it has unburdened me … Yes, I trust them. (Oskar)

#### 3.2.2. Inclusion

The next of kin described that they experienced great support from the staff in the MGT as they were someone you could turn to in all situations and ask questions and obtain advice, help and answers from. The next of kin also described that they often had felt left out and had not been included in the medical decisions regarding the older adult. However, the staff from the MGT took the time to include them and explain to the next of kin what was going on with the older adult’s health. The next of kin thought it was very important that they could get involved and feel included in the care. They believed that the MGT had a conscious strategy for this, and some next of kin felt that they were like one of the team, that they were involved, and that the communication worked well between them and the staff. The next of kin highlighted aspects of the MGT’s working methods, such as proximity, planning and feedback. Several next of kin described that direct feedback was important, and they felt very calm and secure in receiving information about how the care is planned for their older family member in need of medical efforts.

Every week they [the MGT] called and gave me an update, so you always knew what was going to happen. … And then you become calm. And I think that’s right, because, with the result in hand … And then you’ve got the whole thing with security and all that, and then when it’s good they do not have to call, because then it’s good after all. (Hans)

Other next of kin did not feel as involved but were still satisfied. Even those who were not as involved felt that they could call the MGT at any time and be listened to.

They inform you, they really do. They are very clear in saying what they do and why … So, I think they have been very clear on that … Other than that, I have not really been involved or influenced anything. I don’t think so. I suppose I have just accepted what they [the MGT] have suggested. Because I suppose I think they have the expertise. (Oskar)

Some next of kin also highlighted that the MGT not only included them in the medical treatment but also took the time to explain and include the older adults themselves. 

#### 3.2.3. Relief

The experience of having a family member who needs extensive care and attention varies among the next of kin. Some of them lived with older adults and experienced more stress than those who lived apart. Several described that it was difficult to think about themselves and their own needs. When the team became involved, the next of kin felt that they received support in their own situation and a sounding board, someone they could turn to in all situations, could ask questions and obtain advice, help and answers from.

Being the family caregiver of a seriously ill older adult was described as a great responsibility and was also sometimes perceived as a burden that could be difficult to bear. They expressed how it had previously affected their everyday lives to be constantly worried about their loved ones and that they were often contacted by them and had to make emergency visits when they were not feeling well.

When they received help from the MGT, they described it as though they were part of a large team and that the team not only cared about the patient, i.e., the older adult but that they were also interested in the next of kin and their situation.

They also ask me how I feel, so that I can cope. It’s a big strain on me, too. It’s a relief for me and I feel I can trust them. They have specialist knowledge in the area of older people—the staff at the health centre does not have that. (Oskar)

So, as a next of kin, I think it’s great not to get these calls that now I’m in the emergency room again. It has taken a lot of time away from my family, my husband and my children; he [my husband] has been very supportive in this. But … It’s nice not to have to feel the stress that I must go to school and then go and visit my dad, come home in the evening, barely see my children. (Karin)

Several next of kin experienced that the entire care situation changed with the MGT. With the MGT, they got someone to communicate with and someone to take responsibility for their loved ones, and they expressed that they felt relieved. The relief that the next of kin experienced through the MGT taking over the responsibility meant that more of them could go away and do their own activities and have their own time to take care of themselves.

That I can go shopping or do some other stuff and so on. Before I did not dare … hardly walked out the door because I was afraid he would fall and so on. (Svea)

Some next of kin described a kind of relief when they saw that the MGT and they themselves had the same intentions as to how the healthcare should be provided for the older adult afflicted by multiple diseases, and how this became a further relief when they realised, they were sharing the same views.

## 4. Discussion

The aim of the study was to describe how next of kin experience care efforts from a mobile geriatric team for their older adult family members. Through the next of kin’s descriptions of their experiences, this study has revealed two descriptive categories: Professional care and No longer having the main responsibility.

The findings show that all next of kin highlighted the MGT staff’s competence in geriatrics. Most of the next of kin experienced a huge difference in competence between primary care and MGT. However, this praise was also expressed along with a rather scathing critique of what was described as a lack of competence and coordination in municipal care and primary care. Several studies highlight that next of kin value staff competence highly. Their competence has great significance for how next of kin experience the quality of care and how satisfied they are with how the staff provides care to the older adult [33,34].

This finding is in line with a survey carried out by the Swedish National Board of Health and Welfare [35]. It appears as though the level of education in geriatrics and gerontology within the university’s basic education in healthcare is too low in all professional categories. There are, thus, too few specialist nurses and specialist physicians with a focus on older adults. In 2021, the regions and municipalities received grants from the state to enable professionals to attend specialist training in geriatrics and gerontology to increase competence in the area [36].

The next of kin’s descriptions of previous experiences reflect a rather fragmented care system where they felt that the continuity of care did not work satisfactorily. However, several studies [37,38] have suggested that a cohesive continuity of care is the basis for good healthcare with high quality, more satisfied patients, better health, and more equal and cost-effective care. In our study, the next of kin state that there is a lack of continuity of care for older adults in the primary healthcare system. However, they emphasise that the continuity was good in the MGT, both when it comes to regular visits and in terms of the same persons from the team visiting the older adult. In line with Robinson et al. [39], continuity appears to be a prerequisite for the staff and next of kin to be able to build a good relationship. 

The next of kin described difficulties with primary care and the health centre before the MGT took over. Often, they had to accompany the older adult to the emergency room, where they had to wait several hours before receiving help. Kirsebom et al. [5] describe this as a known problem; multi-sick older adults are sent between different specialities and levels of care, which means that no one in the care chain takes a holistic approach to the patient’s problems.

All next of kin expressed that the most rewarding thing about the MGT was that the physician made home visits. Having a physician within the MGT also meant that they had the power to make decisions much faster. Here it should perhaps be clarified for the reader that home visits from physicians are very unusual in Sweden because the doctors are employed by primary care and not by municipal health and medical care. The Swedish National Board of Health and Welfare [40] has criticised the lack of home visits from physicians in a report. Particularly, the board highlighted that patient safety needs to be developed in connection with care transitions, such as enrolment in and discharge from closed specialised care for older adults. In this way, the MGT is also quite unique for Swedish healthcare, with its collective competence that also includes physicians and the team continuously coming to the same patients. Ellis and Sedvalis [9] concluded that the MGT in their study could manage patients safely in their own homes during clinical crises, avoiding unnecessary and often detrimental lengthy stays in acute hospitals. That treatment and care took place at home was also important for the next of kin in our study, who otherwise had to accompany their older adult family member to different clinics and specialists. If necessary, other specialities in healthcare could also be contacted by the MGT in an informal way to solve the patient’s problems without a lot of waiting time. Earlier research has shown that coordinated care can result in fewer unplanned hospital admissions [41].

A much-appreciated effect of the interprofessional approach in the MGT was the advanced medical treatment that was given within the older adult’s home. It is normally not possible to get, e.g., intravenous treatments or blood transfusions at home. With the efforts from the MGT, hospitalisation could be avoided. The next of kin described this as a very important component in the team’s working methods. Condelius et al. [6] show in their study that receiving treatment at home reduces stress for next of kin as they do not have to go to the hospital for treatment. 

In Sweden, next of kin provide most of the care in the home for older adults [2]. Research shows that many next of kin experience stress and demands from their surroundings to be available for their older family member [9,42]. In this study, the next of kin experienced that they had to take great responsibility for the older adult, and as a consequence, they found it difficult to prioritise their own needs, especially if living together. Yet, the next of kin described feelings of reduced responsibility for the older adults in need of care when the MGT came into the picture. 

In this study, the next of kin talked about experiencing great stress in everyday life, to such an extent that they were close to a breakdown and getting sick themselves. Several stated that they would not have been able to cope anymore if their older family members had not been offered help from the MGT. The next of kin stated that they received support from the MGT and that they were good conversation partners. This is in line with Törnfelt et al.’s study [17], which points out that the interpersonal factors between the staff and the next of kin are important for creating security and quality of care for older adults and the next of kin. 

During visits to the hospital or health centre, the next of kin felt invisible instead, as they, for example, were not included in the medical decisions regarding the older adult. In contrast, the MGT staff took the time to include the next of kin and explain to them what was going on with the older adult’s health. Ekman et al. [43] show in their study that when the care staff takes the time to listen to the patient and their next of kin, the effects are a shorter care time and higher satisfaction for the patient. Ryan et al. [44] describe the importance of the staff creating a partner-like relationship between those involved, which is crucial for the well-being of the older person and next of kin. This is necessary so that the staff’s and the next of kin’s resources can be maximised to the benefit of all involved.

In our study, the next of kin also expressed that they had been given more freedom and could start to think about themselves and their own needs. Trusting the MGT to be able to take full responsibility was experienced as a relief. It also made it easier for the next of kin to manage their own lives without having to feel that they always needed to be on standby in case of an emergency. This is in line with Wallroth [16] showing that when medical institutions are lacking or when next of kin do not trust those who provide the care or medical efforts, it causes a lot of anxiety and stress. 

### 4.1. Methodological Considerations

To ensure rigour in the method, several core components of qualitative research were considered [45]. The research group had different professional competencies—that of a social worker, a lecturer in social work and a registered nurse—which may strengthen the trustworthiness of the study. In all studies, there is, of course, a risk that the researchers’ own pre-understandings colour the results of the study. In this study, two of the researchers are grounded in the municipal perspective, and one has experience in regional healthcare. It can be seen as a strength that both perspectives are present among the researchers in this study. The same person, an experienced social worker, conducted all the interviews. This may have meant that the participants felt a sense of confidence based on the interviewer’s competence, perhaps reflected in open and trustful answers. However, there can be a risk of answers being designed to fit what the person thinks the interviewer wants to hear. An effort was made to avoid socially desirable responses by asking follow-up questions so the interviewees could be reflective and speak freely about how they experienced the MGT’s healthcare efforts. None of the next of kin had a dependent relationship with the interviewer that could have influenced the results.

A variation in the next of kin’s gender, age, experience of care and relationship to the older adult gave a variety of conceptions, which strengthened the credibility of the study. To increase the credibility of the findings, an open-ended interview guide was used in all interviews to ensure that the same questions were asked and to avoid too much variation. All next of kin in this study gave detailed and varied descriptions. Dependability was ensured by providing a careful description of the method. Phenomenography seeks to understand, describe and conceptualise the variation of people’s conceptions of the world around them [25,27]. Data were analysed in line with the works of Marton [27] and Marton, Dall’ Alba and Beaty [28], identifying patterns of similarity and dissimilarity. The researchers read the data, cross-checked all steps of the analysis and reflected critically on the concepts and descriptive categories. Quotations were used in order to illuminate the findings and strengthen confirmability [46]. The transferability [47] of the results to other groups of next of kin with a family member receiving help from the MGT in other countries and contexts is uncertain, as the study was set in Sweden, and only fourteen next of kin participated.

### 4.2. Strengths and Limitations 

One strength of this study is that it examines the experience of MGT from a next-of-kin perspective, a perspective that only a few studies have taken. Among the participants, there were more women than men; this can be seen as a limitation, as men’s perspective in caregiving often is lacking [16]. Another limitation is that those who chose to participate in the study perhaps were not the ones with the most extensive care needs. 

### 4.3. Relevance to Clinical Practice 

The MGT approach contributes to a cohesive care chain for the older adult patient and a relief for the next of kin. This study shows that an important part of the working method is to involve next of kin in the older adult’s medical care. The next of kin should be involved in the care and be given support so that their burden is reduced, and they can feel confident and secure. The MGT’s approach to involving next of kin means that they are seen as a resource with important competencies and life experience. This will be a challenge for the care staff in the MGT. One conclusion from our study is that the entire healthcare system needs to learn to work together. Thus, it is important to take on a holistic, multidimensional view when managing weaknesses and dealing with clinical challenges. The results from our study contribute to knowledge on how to meet the next of kin’s needs and provide a basis for further development of the MGT initiative. This knowledge can also be used in the education of care staff and in quality improvement work. 

## 5. Conclusions

The MGT initiative was clearly appreciated and valued highly among the next of kin. However, their satisfaction with the MGT must also be seen as a rather harsh critique of the way care and medical efforts are organised (or rather unorganised) when it comes to older adults and elder care. Our study shows that the next of kin experience relief from the burden of responsibility for the older adults. The next of kin experienced the MGT’s physicians and nurses as much more professional and more aware and knowledgeable about the older adults’ problems and disease pictures than those working within the general healthcare system. What was most appreciated was that the physicians and nurses made home visits and that the care and treatment took place at home, which became a relief for the next of kin as they otherwise would have had to accompany the older adult to a health centre or hospital.

Overall, the findings show that well-functioning collaborations, including frequent and continuous communication between team members within the team, between the team members and patients and their next of kin, and between team members and other caregivers, were considered to be fundamentally important facilitators for a high quality of care. This knowledge can be used in the improvement of the MGT initiative and constitutes important information for the care staff in their work. In general, more studies are needed that focus on older adults’ needs and wishes concerning how they want their medical treatments and care efforts to be organised and include the perspective of their next of kin. More specifically, there is a need for more research on the older adults’ own experiences of receiving care from the MGT and a need for extended studies that include a larger number of participants who have experience of being next of kin of older adults who receive care from an MGT. 

## Figures and Tables

**Table 1 healthcare-11-01038-t001:** Information about the participants.

Name	Sex	Age	Relationship to the Older Person
Daniel	Male	62	Son
Karin	Female	27	Daughter
Helena	Female	46	Daughter
Eva	Female	73	Fiancée
Birgit	Female	59	Daughter-in-law
Hans	Male	52	Son
Maria	Female	71	Sister
Gunnar	Male	82	Husband
Oskar	Male	63	Son
Anna	Female	63	Daughter
Ingrid	Female	73	Wife
Svea	Female	73	Wife
Stina	Female	75	Wife
Valter	Male	36	Son

**Table 2 healthcare-11-01038-t002:** Descriptive categories and conceptions.

Descriptive Categories	Professional Care			No Longer Having the Main Responsibility		
**Conceptions**	Competent staff	Person-centred care	Increased availability	Trust	Inclusion	Relief

## Data Availability

The data that support the findings of this study are available from the corresponding author upon reasonable request.

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
