# Peer review of "Efforts of a Mobile Geriatric Team from a Next-of-Kin Perspective: A Phenomenographic Study"

_healthcare, 2023, doi:10.3390/healthcare11071038_

Round 1

Reviewer 1 Report

Thank you for the opportunity to review this manuscript. This study had a relevant aim which is under is underexplored related to older people and the care they receive. The Mobile Geriatric Team has been an available option for many years, but the focus in the research is usually related to the team and the patients care which limits their need for hospitalizations, and saves money for society.

However in the current form the manuscript is not ready for publishing. Please consider the comments below to support you in improving this manuscript which has potential and will shine light on the next of kin, which appears to be undervalued when caring for their older family members.    

Introduction:

Please add a reference for this " Contrary to many people’s beliefs, it is not public care for older adults that provides the most care in Sweden. Instead, the next of kin of those in need provide most of the care"

 It sounds like MGT is founded on/taken from the Comprehensive Geriatric Assessment (CGA) , might want to add this as a reference... I know in Sweden qualitative research has been done on the patients receiving CGA, where they experienced feeling respected as a person, but no mention of the next of kin, which supports you identifying a gap in the research. Reference: Geriatrics (Basel), 2019 Jan 25;4(1):16. doi: 10.3390/geriatrics4010016. "(MGT) ambition is to take a holistic approach and see the person in need of care as a whole, considering medical, social, psychological, and physical aspects, as well as taking the perspective of the next of kin into account".

2.5 Data analysis: did you use a software ie NVivo to support you in your data analysis? If not, should consider adding this as a weakness to your study (4.2 strength and weakness), since we are humans and the software can identify things that may have been overlooked, greater chance that bias is not as prevalent. 

Table 2. Descriptive categories and conceptions. (it would appear that the authors have not exhausted their analysis and categorizations yet. There are  still many overlapping phalanges, which make it appear that another further analysis of the latent experiences could be further condensed. For example the descriptive categories are all related to the fact that the MGT gives and is Professional care, which will result in and could suggest that the next of kin  No longer having the main responsibility and that the MGT Facilitated  everyday life. Results emerge from the data and are not imposed by the authors. Just a suggestion that the authors secure the categories and conceptions so they can stand on their own and do not overlap, while making sure the generalizations about the particular phenomenon are clearly perceived or experienced with participants quotations/examples to support the data.

For example this quote is from facilitating life, but could fit under competent staff as well. These things should be clear for the readers so they are not second guessing or placing things into different categories as I find myself doing, while reading your manuscript... 3.3.1 "They have specialist knowledge in the area of older people – the staff at the health centre does not have that. (Oskar)

OR 3.2.2 They address her very nicely and have a dialogue and they talk to her, not over her head, and I appreciate that as a next of kin. And that’s how it should be … I think she’s calmer. (Birgit) could be moved to 3.1.2 Holistic view...or could be renamed to person centred care, since holistic care is probably the authors bias and the categorization is named after the MGT ambition from your background.   

Furthermore, The manuscript would be easier to read if the authors limited the number of quotations and example for each sub-category/conception and merely used only one. Recommend killing your darlings and keep the best which supports and illuminates what you want your readers to know. In the current format, the manuscript it is too cluttered and the reader loses interest.

Recommend remove the term "elderly" from the entire manuscript. This term is considered discriminatory. Consider replacing with older people or in this example with geriatrics.  "There are thus too few specialist nurses and specialist physicians with a focus on the elderly. 

Consider discussing author bias in 4.1 methodology or in strengths and limitations (4.2) . Despite good intentions, this is very difficult to avoid when doing qualitative research.  

Reviewer 2 Report

This article explains in a qualitative way the advantages of Mobile Geriatric Teams (MGTs)

11)      Section1: “Therefore, the aim of this study is to describe how next of kin experience care efforts from a mobile geriatric team for their older adult family members.”

Could you explain if your study is different from others? In this case, explain why this difference is important. If it is not different, explain if your results are more significant.

22)      Could you add some quantitative analysis? For example, compare the economic cost between the usual system and the use of MGT in your group of patients? Is the cost for the health system higher? Is the cost to patients lower?

33)      You have explained the use of MGT with 14 people. Do you know the maximum number of people that your MGT team could manage without losing quality?

44)      Today technology is helping older people to stay at home as long as possible. However, not all technology is accepted. What kind of technology could be used that could facilitate the work of MGT and that would be well accepted by older people?

Round 2

Reviewer 1 Report

The manuscript has been improved and in my opinion is now eligible for publication.